# Epidemiology of tobacco use in Qatar: Prevalence and its associated factors

**Ahmad AlMulla**[1], **Ravinder Mamtani**[2], **Sohaila Cheema**[2], **Patrick Maisonneuve**[3], **Jamal Abdullah BaSuhai**[1], **Gafar Mahmoud**[1], **Silva Kouyoumjian**[1] *

**1** Department of Medicine, Tobacco Control Center, WHO Collaborative Center, Hamad Medical Corporation, Doha, Qatar, **2** Institute for Population Health, Weill Cornell Medicine-Qatar, Doha, Qatar, **3** Unit of Clinical Epidemiology, IEO Istituto Europeo di Oncologia IRCCS, Milan, Italy

* SKouyoumjian@hamad.qa

**Data Availability Statement:** All relevant data are within the paper and its Supporting Information files.

## Abstract

Tobacco use is a serious public health concern as it causes various deleterious health problems. The aim of this study was to determine the prevalence of tobacco use and various types of tobacco used among a population-based sample of adults 18 years and above in Qatar (residents and expatriates). The study also attempted to assess tobacco use initiation age, tobacco dependency, and to identify factors associated with current tobacco use. This 2019 cross-sectional study was conducted among governmental employees and University students in Qatar using cluster sampling methodology. Study participants completed a self-administered, country-adapted summarized version of the Global Adult Tobacco Survey. 25.2% (n = 1741; N = 6904) of the surveyed sample reported current tobacco use. 21.5% (n = 1481) smoked tobacco (cigarettes, waterpipe, medwakh and cigar) concomitant with other forms of tobacco and only 1.0% (n = 69) were using other forms of tobacco (electronic cigarettes, smokeless tobacco and heat-not-burn tobacco products) and 2.7% (n = 191) did not mention the type of tobacco products used by them. Of the 1550 tobacco users, 42.8% were cigarette smokers, 20.9% waterpipe, 3.2% *medwakh* (Arabic traditional pipe) and 0.7% cigar. Moreover, 1.9% reported smokeless tobacco use (*sweika*), 2.0% electronic cigarette use, and 0.3% heat-not-burn tobacco use. The mean age for smoking initiation was 19.7±5.3 (Qataris 18.6±4.8 and non-Qataris 20.3±5.6). Using multivariable logistic regression, significant association was observed between tobacco use and gender, nationality, age, monthly income, living with a smoker, and self-rated health. This large population-based cross-sectional survey provides the first evidence for the prevalence of different types of tobacco use including *medwakh* smoking among adults (Qataris and non-Qataris) 18 years and above in Qatar. This can serve as a baseline for future research studies on the topic. Based on the review of previous and current tobacco survey findings, it is evident that the prevalence of tobacco use (current) in Qatar has declined suggesting that tobacco control measures implemented by the country have been effective in reducing tobacco consumption.

**Funding:** Ahmad AlMulla AA received funding. This work was supported by the Medical Research Center in Hamad Medical Corporation Doha-Qatar (MRC- Routine grant 01-17-073), https://www.hamad.qa/EN/Education-and-research/Medical_Research/Pages/default.aspx. The findings achieved herein are solely the responsibility of the authors.

**Competing interests:** None to declare.

## Introduction

Smoking remains one of the leading risk factors causing premature death and disability worldwide [1]. Though tobacco use in the World Health Organization (WHO)'s Eastern Mediterranean Region (EMR) is projected to have slightly decreased [2], yet the number of smokers are significant [3] and cigarette affordability remains the highest as compared to the rest of the world [4]. The smoking prevalence varies among the EMR countries; in general, it reaches up to 50% among men and around 10% among women [5]. Tobacco smoking is gaining widespread acceptance among the young across the region with the use of various tobacco products [6–8]. Emerging trends of vaping and the use of tobacco devices [9], which may lead to a higher disease and economic burden are of serious concern [10].

Qatar has adopted the WHO's Framework Convention on Tobacco Control. In line with the Qatar National Vision 2030 and the Ministry of Public Health Strategy 2018–2022, Qatar seeks to curb the current smoking prevalence by 5%. The Ministry conducted the Global Adult Tobacco Survey (GATS) in 2013, which reported that 12.1% of 15 years old and above currently smoked tobacco [11]. According to a previous study conducted by Al-Mulla et al, the current prevalence of cigarette smoking in Qatar was reported as 36.5% [12]. In this study, the Tobacco Control Center, WHO Collaborating Center, Hamad Medical Corporation, Qatar aimed to update the status of tobacco epidemiology in the country using the methodology previously utilized in the year 2000 [12]. Our study objectives were as follows: 1) determine the prevalence of tobacco use among a population-based sample of adults 18 years old and above in Qatar (residents and expatriates), 2) determine the different types of tobacco used, 3) assess tobacco dependency, age and reasons of initiation for current tobacco use, and 4) identify factors associated with tobacco use.

The study findings will be instructive to promote an improved understanding of tobacco use epidemiology, and provide the necessary evidence to guide future research, policy-making decisions, and programmatic intervention strategies for future tobacco control in Qatar.

## Materials and methods

### Study design, setting and sample

A cross-sectional study was conducted among governmental employees and university students in the state of Qatar (S1 Fig in S1 File). The students were selected from four governmental universities namely, Qatar University, College of North Atlantic in Qatar, Police Academy, and the National Service Academy. The governmental employees were selected from the health sector, 12 ministries and 10 governmental authorities and units. The rationale for selecting this population was to have a fair representation of Qatari adults, since almost all Qataris prefer to work for the government and receive education in governmental colleges/universities. To be eligible to participate in the self-administered anonymous survey, participants had to be 18 years old or above. Data was collected from March—December 2019. The study's research design and implementation approach were based on a systematic process with due attention given to reduce bias at each stage of the study. The field work was exemplary across all sample recruitment sites to ensure that the results were replicable and valid to support future decision-making for tobacco control.

The study protocol was approved by the Institutional Review Board, Medical Research Center, Hamad Medical Corporation, Qatar.

### Sample size

The sample size (SS) equation in cluster design SS = $[t^{2*} (p^*q/d^2)]$ * DEFF was used to estimate the true proportion of tobacco smoking in Qatar, with a confidence interval of 95% for cluster

sampling, a t-score of 2.045, and an absolute desired precision of d = 0.0175. Calculations were done using the most conservative hypothesized proportion (p = 0.50) and expected non-prevalence of 0.50 (q = 1-p). We used 2.0 for the design effect (DEFF) as a "correction factor" to account for the heterogeneity among the clusters. To account for a non-response of 20%, a SS of 7,784 was required. Hence, the rounded up target sample size was 8000.

## Sampling

We applied multi-level cluster selection before the final sample elements were reached. Regarding the sampling of universities, local colleges were randomly chosen and within these institutions, departments and then classes were randomly selected. All the students within these classes were included in the study.

For the governmental employees, the number of floors within the workplace setting that is building(s) of each entity was added up. These floors were considered as the clusters. Only odd floor numbers were selected for inclusion and all the employees on those floors were included in the study. Twenty percent of the uppermost floors were discarded in the Ministries, considering these floors included employees with higher ranking positions in order not to disturb the daily activities of the minister and its deputy associates.

For instance, if a tower had a total of 50 floors. We considered the 50 floors as 50 clusters of units. After 20% omission, the twenty odd floors were selected for inclusion. All employees on those floors were approached for the study. If the institution had more than one tower/building, the same methodology was applied to each building independently. The sampling was systematically applied throughout the field. Institutions/units that did not agree to comply with our sampling strategy were excluded from the sample.

## Instrument and measures

A self-administered survey was developed, which is a country-adapted summarized version of the Global Adult Tobacco Survey (GATS). The survey was reviewed by WHO Regional Office for the Eastern Mediterranean and was approved contingent upon amending it to include additional questions on heat-not-burn products as advsised by the Global Tobacco Surveillance System (GTSS) Team in Centers for Disease Prevention and Control (CDC). This would allow the amended survey to be consistent with the latest version of the GATS. Additionally, questions about medwakh smoking (Arabic traditional pipe) were incorporated. The survey was piloted among members of the TCC and found to be suitable. The survey tool is available in Arabic and English and is provided as Supporting Information (S1 and S2 Appendixs in S1 File).

The survey instrument was carefully administered with attention to protocol adherence by trained staff to guarantee a unified procedure. Study participants were provided with an envelope that included the survey along with the information sheet describing the study. To ensure anonymity and confidentiality of responses, the respondents were asked to complete the survey and return it via a sealed envelope. Completion of the survey was considered as informed consent by the participant to be included in the study. It took approximately 15–20 minutes to complete the survey.

## Data analysis

Collected data was coded and entered into Statistical Package for Social Sciences (IBM SPSS statistics; version 26; Armonk, NY: IBM Corporation program). Fifty percent of data entry was reviewed and repeated by a different person in order to verify and validate the accuracy of the process. The main outcome variable was the prevalence of current tobacco use expressed as a

proportion (%). The status of tobacco use was divided into two categories: current users and non-users (ex-smokers and never users). Descriptive categorical variables were expressed as proportion while descriptive continuous variables were expressed as mean ± standard deviation (SD). Statistical analysis for the association of various variables with tobacco use was carried out using Chi square test using a significance level of less than 5%. Variables that showed significant association with tobacco use in univariate analysis were entered into multivariable logistic regression to find significant predictors of tobacco use.

## Results

### Sample characteristics

The final sample size was 7921, with an overall survey response rate of 89.7%. Out of 7105 surveys collected, 6904 were complete and included in the analysis. Socio-demographic characteristics of current tobacco users and non-users are shown in Table 1. The majority (n = 1882; 31.4%) of the partcicipants were in the age category of (25–34 years). Of the sample 58.4% (n = 4002) were men, 59.6% (n = 4072) were married, and 47.6% (n = 3263) were of Qatari nationality.

Most of the study participants perceived themselves having excellent (41.2%, n = 2749) and good (53.7%, n = 3588) health. Health problems existed among 38.1% (n = 2453) of the sample (S1 Table in S1 File). Most common morbidities reported by the participants were obesity 12.2% (n = 783), high cholesterol levels 8.5% (n = 548), high blood pressure 7.9% (n = 510) and diabetes 7.1% (n = 454). Additional details of the number and percentage of different types of comorbidities can be found in (S2 Table in S1 File). Approximately half of the sample had a smoker living with them (47.0%) (n = 3195) (S3 Table in S1 File). 26.6% (n = 1805) reported that the brother smoked and 16.8% (n = 1139) mentioned that the father smoked (S4 Table in S1 File).

### Prevalence of current tobacco users versus non-users

Overall, 25.2% (n = 1741) of the sample were current tobacco users, 12.0% (n = 828) were ex-tobacco users and 62.8% (n = 4335) were never users (Table 1 and S5 Table in S1 File). Of the total sample, 36.6% (n = 1464) male and 9.2% (n = 263) of females were current tobacco users. Males had significantly higher odds of being tobacco users than females [OR = 5.68, 95% CI 4.93–6.55]. Current tobacco use was higher among non-Qataris 29.3% (n = 1049) in comparison to Qataris 20.6% (n = 671). Non-Qataris had significantly higher odds of being a tobacco user than Qataris [OR = 1.60, 95% CI 1.43–1.79]. The highest prevalence of tobacco use was observed among participants aged 18–24 (27.2%; n = 367). The age category of ≥55 years had significantly lower odds [OR = 0.59, 95% CI 0.42–0.83] of being tobacco users compared to the age category 45–54 years. Individuals with a monthly income of 5000–9999 QAR had significantly lower odds [OR = 1.28, 95% CI 1.04–1.571] of being a tobacco user than individuals with a higher monthly income of >40000 QAR. Persons who rated themselves as having excellent health had lower odds [OR = 0.43, 95% CI 0.34–0.55] of being a tobacco user than those who rated themselves as having good health. Individuals who mentioned that they are living with a tobacco user had higher odds [OR = 1.60, 95% CI 1.44–1.79] of being a tobacco user than individuals who did not live with a tobacco user. No significant difference in frequency of tobacco use was found between individuals of different marital status and education.

### Prevalence of current tobacco use by nationality

Table 2 shows the prevalence of current tobacco use by Qataris and non-Qataris. Current tobacco use is predominant among Qatari males (42.1%) in comparison to Qatari females

**Table 1. Sociodemographic characteristics of current tobacco users and non-users.**

| | Total^ | Current tobacco users* | Non-tobacco users | p value |
|---|---|---|---|---|
| | (n, % of total) | (n, %) | (n, %) | |
| All subjects | 6904 (100%) | 1741 (25.2%) | 5163 (74.8%) | |
| **Age** | | | | |
| 18–24 | 1351 (22.5%) | 367 (27.2%) | 984 (72.8%) | 0.007 |
| 25–34 | 1882 (31.4%) | 503 (26.7%) | 1379 (73.3%) | |
| 35–44 | 1692 (28.2%) | 441 (26.1%) | 1251 (73.9%) | |
| 45–54 | 811 (13.5%) | 186 (22.9%) | 625 (77.1%) | |
| 55+ | 266 (4.4%) | 48 (18.0%) | 218 (82.0%) | |
| **Gender** | | | | |
| Male | 4002 (58.4%) | 1464 (36.6%) | 2538 (63.4%) | <0.0001 |
| Female | 2854 (41.6%) | 263 (9.2%) | 2591 (90.8%) | |
| **Nationality** | | | | |
| Qatari | 3263 (47.6%) | 671 (20.6%) | 2592 (79.4%) | <0.0001 |
| Non-Qatari | 3585 (52.4%) | 1049 (29.3%) | 2536 (70.7%) | |
| **Marital Status** | | | | |
| Single | 2555 (37.4%) | 651 (25.5%) | 1904 (74.5%) | 0.10 |
| Married | 4072 (59.6%) | 1035 (25.4%) | 3037 (74.6%) | |
| Separated/Divorced | 170 (2.5%) | 34 (20.0%) | 136 (80.0%) | |
| Widowed | 35 (0.5%) | 4 (11.4%) | 31 (88.6%) | |
| **Education** | | | | |
| Secondary or less | 1418 (20.8%) | 336 (23.7%) | 1082 (76.3%) | 0.35 |
| University | 4240 (62.1%) | 1085 (25.6%) | 3155 (74.4%) | |
| Postgraduate | 1167 (17.1%) | 289 (24.8%) | 878 (75.2%) | |
| **Profession** | | | | |
| Student | 1188 (17.3%) | 329 (27.7%) | 859 (72.3%) | 0.027 |
| Employee | 5679 (82.7%) | 1399 (24.6%) | 4280 (75.4%) | |
| **Monthly income** | | | | |
| 0–4999 | 806 (13.1%) | 197 (24.4%) | 609 (75.6%) | <0.0001 |
| 5000–9999 | 1051 (17.0%) | 307 (29.2%) | 744 (70.8%) | |
| 10000–24999 | 2565 (41.6%) | 648 (25.3%) | 1917 (74.7%) | |
| 25000–39999 | 1141 (18.5%) | 250 (21.9%) | 891 (78.1%) | |
| 40000+ | 605 (9.8%) | 188 (31.1%) | 417 (68.9%) | |
| **Self-rated health** | | | | |
| Excellent | 2749 (41.2%) | 546 (19.9%) | 2203 (80.1%) | <0.0001 |
| Good | 3588 (53.7%) | 1023 (28.5%) | 2565 (71.5%) | |
| Fair | 339 (5.1%) | 124 (36.6%) | 215 (63.4%) | |
| **Health problems** | | | | |
| Yes | 2433 (38.1%) | 626 (25.7%) | 1807 (74.3%) | 0.65 |
| No | 3949 (61.9%) | 996 (25.2%) | 2953 (74.8%) | |
| **Smoker at Home** | | | | |
| Yes | 3169 (47.0%) | 942 (29.7%) | 2227 (70.3%) | <0.0001 |
| No | 3570 (53.0%) | 745 (20.9%) | 2825 (79.1%) | |

*Includes daily and occasional users.

^Total does not add up due to missing data.

**Table 2. Sociodemographic characteristics of Qatari and non-Qatari current tobacco users.**

| | Qataris | | | Non-Qataris | | |
|---|---|---|---|---|---|---|
| | Total ^ | Current tobacco users* | p value | Total ^ | Current tobacco users* | p value |
| | (n, % of total) | (n, %) | | (n, % of total) | (n, %) | |
| All subjects | 3263 (100%) | 671 (20.6%) | | 3585 (100%) | 1049 (29.3%) | |
| Gender | | | | | | |
| Male | 1460 (44.9%) | 614 (42.1%) | <0.0001 | 2517 (70.5%) | 837 (33.3%) | <0.0001 |
| Female | 1790 (55.1%) | 54 (3.0%) | | 1052 (29.5%) | 207 (19.7%) | |
| Age | | | | | | |
| 15–24 | 766 (27.8%) | 185 (24.2%) | 0.042 | 580 (18.0%) | 180 (31.0%) | <0.0001 |
| 25–34 | 961 (34.9%) | 179 (18.6%) | | 915 (28.4%) | 321 (35.1%) | |
| 35–44 | 670 (24.3%) | 128 (19.1%) | | 1016 (31.5%) | 310 (30.5%) | |
| 45–54 | 298 (10.8%) | 64 (21.5%) | | 508 (15.8%) | 120 (23.6%) | |
| 55+ | 60 (2.2%) | 15 (25.0%) | | 205 (6.4%) | 32 (15.6%) | |
| Marital Status | | | | | | |
| Single | 1447 (44.7%) | 289 (20.0%) | 0.082 | 1098 (30.9%) | 357 (32.5%) | 0.022 |
| Married | 1653 (51.0%) | 358 (21.7%) | | 2395 (67.3%) | 667 (27.8%) | |
| Separated/Divorced | 117 (3.6%) | 16 (13.7%) | | 53 (1.5%) | 18 (34.0%) | |
| Widowed | 22 (0.7%) | 2 (9.1%) | | 13 (0.4%) | 2 (15.4%) | |
| Education | | | | | | |
| Secondary or less | 981 (30.4%) | 206 (21.0%) | 0.51 | 426 (11.9%) | 126 (29.6%) | 0.003 |
| University | 2064 (63.9%) | 410 (19.9%) | | 2159 (60.6%) | 668 (30.9%) | |
| Postgraduate | 187 (5.8%) | 43 (23.0%) | | 975 (27.4%) | 244 (25.0%) | |
| Profession | | | | | | |
| Student | 617 (19.0%) | 156 (25.3%) | 0.001 | 567 (15.9%) | 171 (30.2%) | 0.59 |
| Employee | 2638 (81.0%) | 512 (19.4%) | | 3010 (84.1%) | 874 (29.0%) | |
| Monthly income | | | | | | |
| 0–4999 | 181 (6.2%) | 37 (20.4%) | <0.0001 | 621 (19.3%) | 158 (25.4%) | 0.092 |
| 5000–9999 | 203 (6.9%) | 37 (18.2%) | | 841 (26.2%) | 265 (31.5%) | |
| 10000–24999 | 1234 (42.2%) | 255 (20.7%) | | 1319 (41.1%) | 389 (29.5%) | |
| 25000–39999 | 821 (28.1%) | 153 (18.6%) | | 316 (9.8%) | 95 (30.1%) | |
| 40000+ | 486 (16.6%) | 148 (30.5%) | | 116 (3.6%) | 40 (34.5%) | |
| Self-rated health | | | | | | |
| Excellent | 1553 (49.2%) | 262 (16.9%) | <0.0001 | 1184 (34.0%) | 282 (23.8%) | <0.0001 |
| Good | 1516 (48.0%) | 364 (24.0%) | | 2051 (58.9%) | 648 (31.6%) | |
| Fair | 88 (2.8%) | 25 (28.4%) | | 250 (7.2%) | 98 (39.2%) | |
| Health problems | | | | | | |
| Yes | 1147 (38.6%) | 242 (21.1%) | 0.38 | 1271 (37.6%) | 375 (29.5%) | 0.45 |
| No | 1825 (61.4%) | 375 (20.5%) | | 2107 (62.4%) | 616 (29.2%) | |
| Smoker at Home | | | | | | |
| Yes | 1828 (57.4%) | 415 (22.7%) | <0.0001 | 1319 (37.5%) | 516 (39.1%) | <0.0001 |
| No | 1356 (42.6%) | 239 (17.6%) | | 2198 (62.5%) | 502 (22.8%) | |

*Includes daily and occasional users.

^Total does not add up due to missing data.

(3.0%) (p-value<0.001) and similarly among non-Qatari males (33.3%) compared to non-Qatari females (19.7%) (p-value <0.001). Among Qataris, tobacco use was relatively higher among age groups 18–24 years old (24.2%) and 55 years and above (25.0%). For non-Qataris it

was highest for the age group 25–34 years old (35.1%). The prevalence of tobacco use was higher among Qatari students (25.3%) than among employees (19.4%) (p-value = 0.001), whereas among the non-Qataris the difference was not significant (p-value = 0.59). For non-Qataris, tobacco use was highest among university graduates 30.9% (p-value = 0.003) and among separated/divorced (34.0%) individuals (p-value = 0.02), whereas these differences were non-significant among Qataris. Among Qataris, more tobacco users were found with monthly incomes of >40,000 QAR (30.5%), while among non-Qataris the difference was not significant (p-value = 0.09). For both Qataris and non-Qataris, the prevalence of tobacco use was higher when there was a smoker at home, 22.7% and 39.1% respectively (p-value<0.001).

## Types of tobacco use

Table 3 shows the number and percentage of the various types of current tobacco used by Qataris and non-Qataris. Of the total surveyed sample, 25.2% (n = 1741) indicated they were current tobacco users. 21.5% (n = 1481) smoked tobacco (cigarettes, waterpipe, medwakh and cigar) concomitant with other forms of tobacco and only 1.0% (n = 69) were using other forms of tobacco (electronic cigarettes, smokeless tobacco and heat-not-burn tobacco products) and 2.7% (n = 191) did not mention the type of tobacco products used by them.

Table 3 describes the number and percentage of different types of tobacco among current users by nationality. The numbers of different types of tobacco use are not exclusive and further details are found in (S6-S9 Tables in S1 File). Of the current tobacco users (n = 1550), 42.8% were cigarette smokers, 20.9% waterpipe, 3.2% *medwakh* (Arabic traditional pipe) and

**Table 3. The number and percentage of different types of current tobacco use by nationality (n, %) smoking refers to tobacco smoking (cigarettes, waterpipe, *medwakh* and cigar).**

| Types of tobacco use | All tobacco users (n = 1550)* | | | Qatari users (n = 617)* | | | Non-Qatari users (n = 912)* | | |
|---|---|---|---|---|---|---|---|---|---|
| | Any use | Exclusive use | Concomittant with other type | Any use | Exclusive use | Concomittant with other type | Any use | Exclusive use | Concomittant with other type |
| | n (%) | n (%) | n (%) | n (%) | n (%) | n (%) | n (%) | n (%) | n (%) |
| **Tobacco smoking** | | | | | | | | | |
| Cigarette | 1031 (66.5) | 664 (42.8) | 367 (23.7) | 400 (64.8) | 224 (36.3) | 176 (28.5) | 619 (67.9) | 434 (47.6) | 185 (20.3) |
| Waterpipe | 575 (37.1) | 324 (20.9) | 251 (16.2) | 235 (38.1) | 112 (18.2) | 123 (19.9) | 329 (36.1) | 205 (22.5) | 124 (13.6) |
| *Medwakh* | 174 (11.2) | 50 (3.2) | 124 (8.0) | 100 (16.2) | 29 (4.7) | 71 (11.5) | 72 (7.9) | 21 (2.3) | 51 (5.6) |
| Cigar | 49 (3.2) | 11 (0.7) | 38 (2.5) | 22 (3.6) | 3 (0.5) | 19 (3.1) | 27 (3.0) | 8 (0.9) | 19 (2.1) |
| **Smokeless tobacco** | | | | | | | | | |
| *Sweika/tambak* | 91 (5.9) | 30 (1.9) | 61 (3.9) | 66 (10.7) | 18 (2.9) | 48 (7.8) | 25 (2.7) | 12 (1.3) | 13 (1.4) |
| **E-cigarette use** | | | | | | | | | |
| E-cigarette | 175 (11.3) | 31 (2.0) | 144 (9.3) | 75 (12.2) | 12 (1.9) | 63 (10.2) | 97 (10.6) | 18 (2.0) | 79 (8.7) |
| **Heat-not-Burn tobacco** | | | | | | | | | |
| Heat-not-Burn tobacco | 47 (3.0) | 5 (0.3) | 42 (2.7) | 28 (4.5) | 4 (0.6) | 24 (3.9) | 19 (2.1) | 1 (0.1) | 18 (2.0) |
| **More than one type** | | | 435 (28.1) | | | 215 (34.8) | | | 213 (23.4) |

*Data on the type of tobacco was missing for 191 subjects (54 qatari and 137 non-qatari).

0.7% cigar smokers. Moreover, 1.9% reported smokeless tobacco use (*sweika*), 2.0% electronic cigarette use, 0.3% heat-not-burn tobacco use, and 28.1% were using more than one type of tobacco product (Table 3).

Among Qatari current tobacco users (n = 617), 36.3% were cigarette smokers, followed by waterpipe smoking (18.2%), *medwakh* smoking (4.7%) and cigar smoking (0.5%). Moreover, 2.9% reported smokeless tobacco use (*sweika*), 1.9% electronic cigarette use, 0.6% heat-not-burn tobacco use and 34.8% were using more than one type of tobacco product (Table 3).

Among non-Qatari tobacco users (n = 912), cigarette smoking was the highest (47.6%), followed by waterpipe smoking (22.5%), *medwakh* smoking (2.3%) and cigar smoking (0.9%). Moreover, 1.3% reported smokeless tobacco use (*sweika*), electronic cigarette use (2.0%), and 0.1% heat-not-burn tobacco use, and 23.4% were using more than one type of tobacco product (Table 3).

## Dependency, age and reasons of initiation of current tobacco use

Table 4 shows the dependency, initiation age and reasons for tobacco use/smoking among Qataris and non-Qataris. Among the current tobacco users, 36.6% had their first tobacco use/ smoke within 0–30 minutes after waking up (40.6% of Qataris and 33.0% non-Qataris). Less than one third of tobacco users smoked 20 cigarettes or more per day (36.4% Qataris and 27.9% non-Qataris). The mean age of tobacco use/smoking initiation age was 19.7±5.3 (Qataris 18.6 ±4.8 and non-Qataris 20.3±5.6). Half of the current tobacco users began to smoke/use tobacco before the age of 18 (60.9% of Qataris and 43.9% of non-Qataris) and 18.8% before the age of 15 (27.5% of Qataris and 13.7% of non-Qataris). The most common brand of cigarette purchased was Marlboro (42.8%, N = 1008), followed by Parliament (15.5%), Davidoff (12.1%), Dunhill (10.5%), Philip Morris (8.7%), Winston (4.2%), Kent (1.3%), and others (7.9%) (S10 Table in S1 File). Among current tobacco users, the most frequent reasons for smoking initiation were social pastime (46.9%), friend/peer pressure (26.3%), stress relief (23.6%), family member influence (8.1%), media influence (3.8) and others (11.2%). Social pastime and stress relief were the most reported reasons aong the Qatari users, while social pastime and friends were the most reported reasons among the non-Qatari users (Table 4).

## Factors associated with tobacco use

Using multivariable logistic regression (Table 5), associations were observed between tobacco use and self-rated health, living with a smoker, monthly income, nationality, gender, and age. Men were 6 times more likely to be current tobacco users compared to women (adjusted odds ratio (AOR) = 6.37, 95% CI 5.36–7.58); non-Qataris were more likely to be current tobacco users than Qataris (AOR 1.60, 95% CI 1.34–1.92) and respondents 55 years old and above were less likely to be tobacco users compared to respondents in the category of 18–24 years old (AOR 0.26, 95% CI 0.17–0.40). Respondents who were living with a smoker at home were 2.21 times more likely to use tobacco than those living without smokers at home (AOR 2.21, 95% CI 1.92–2.54). Respondents with a monthly household income of more than 40000 QAR were almost three times more likely (AOR 2.95, 95% CI 2.03–4.27) to use tobacco than those with a monthly income of <5000 QAR. Those who had an excellent and good self-rated health status were less likely to use tobacco than those who rated their health as poor (AOR 0.39, 95% CI 0.29–0.53 and AOR 0.67, 95% CI 0.50–0.89, respectively).

## Discussion

In this study, the prevalence of current tobacco use among adults (Qataris and non-Qataris) is 25.2% and tobacco smoking is 21.5% significantly lower than the previous study (36.7%) [12].

**Table 4. Dependency, age and reasons of initiation of current tobacco users by nationality (n, %).**

| | Total^ | Qatari | Non-Qatari |
|---|---|---|---|
| **Dependency** | | | |
| | N = 1152 | N = 510 | N = 630 |
| First smoke/use within 0–30 minutes waking up | 422 (36.6%) | 207 (40.6%) | 208 (33.0%) |
| First smoke/use within 31–60 minutes waking up | 270 (23.4%) | 130 (25.5%) | 138 (21.9%) |
| First smoke/use within >60 minutes waking up | 460 (40.0%) | 173 (33.9%) | 284 (45.1%) |
| | N = 862 | N = 308 | N = 544 |
| Average number of cigarettes per day | 11.5±10.8 | 12.9±10.5 | 10.8±10.9 |
| <1 cig/day | 115 (13.3%) | 28 (9.1%) | 84 (15.4%) |
| 1–9 cig/day | 311 (36.1%) | 103 (33.4%) | 206 (37.9%) |
| 10–19 cig/day | 170 (19.7%) | 65 (21.1%) | 102 (18.8%) |
| 20+ cig/day | 266 (30.9%) | 112 (36.4%) | 152 (27.9%) |
| Missing | 169 | 92 | 75 |
| **Initiation Age** | | | |
| | N = 1560 | N = 601 | N = 944 |
| Average age started smoking/using tobacco | 19.7±5.3 | 18.6±4.8 | 20.3±5.6 |
| | N = 1560 | N = 601 | N = 944 |
| Started smoking/using tobacco ≤15 years | 294 (18.8%) | 165 (27.5%) | 129 (13.7%) |
| Started smoking/using tobacco 16–18 years | 494 (31.7%) | 201 (33.4%) | 285 (30.2%) |
| Started smoking/using tobacco >18 years | 772 (49.5%) | 235 (39.1%) | 530 (56.1%) |
| **Initiation Reasons*** | | | |
| | N = 1573 | N = 605 | N = 950 |
| Family members influence | 127 (8.1%) | 63 (10.4%) | 61 (6.4%) |
| Stress relief | 372 (23.6%) | 129 (21.3%) | 239 (25.2%) |
| Friends/Peer pressure | 414 (26.3%) | 115 (19.0%) | 296 (31.2%) |
| Social/Funtime | 738 (46.9%) | 298 (49.3%) | 432 (45.5%) |
| Media influence | 60 (3.8%) | 22 (3.6%) | 37 (3.9%) |
| Others, specify | 176 (11.2%) | 73 (12.1%) | 102 (10.7%) |

* The same subject may be listed in more than 1 category.

^Total does not add up due to missing data.

However, higher than the rate of Global Adult Tobacco Survey (GATS) in 2013, which reported that 12.1% of 15 years old and above currently smoked tobacco [11]. In our study, tobacco smoking was higher than the smoking prevalence rate reported in the Sultanate of Oman Steps survey (7.1%) [13], however comparable to tobacco smoking rate reported in Kuwait (20.5%) [14], Saudi Arabia (21.4%) [15], and Iraq (20.7%) [16], and much lower than the smoking prevalence rate reported in Lebanon 34.7% [17].

Our study showed that the prevalence of tobacco smoking in Qatar has dropped by 15.2% between 2000 and 2019. This observed decrease is significant. A recent study of healthcare professionals in Qatar showed a relatively lower prevalence of tobacco use than earlier studies [18]. It is evident that Qatar has been successful in reducing smoking prevalence due to a variety of reasons which include: adoption of the WHO's Framework Convention on Tobacco Control, implementation of the 2002 anti-tobacco law Qatar, [19] and other programs in line with the Qatar National Vision of 2030 [20], Qatar Ministry of Public Health Startegy of 2018–2022 and efforts of Tobacco Control Center, WHO Collaborative Center to combat tobacco use.

**Table 5. Logistic regression for current tobacco users by participants' characteristics for the total sample (N = 6904).**

| Characteristic | Tobacco use | p-value[*] |
| --- | --- | --- |
| | Adjusted Odds Ratio (AOR) (95% CI) | |
| Age | | |
| 18–24 | 1.00 (Ref.) | - |
| 25–34 | 0.90 (0.69–1.17) | 0.43 |
| 35–44 | 0.64 (0.48–0.84) | 0.002 |
| 45–54 | 0.43 (0.31–0.59) | <0.0001 |
| 55+ | 0.26 (0.17–0.40) | <0.0001 |
| Gender | | |
| Female | 1.00 (Ref.) | - |
| Male | 6.37 (5.36–7.58) | <0.0001 |
| Nationality | | |
| Qatari | 1.00 (Ref.) | - |
| Non-Qatari | 1.60 (1.34–1.92) | <0.0001 |
| Profession | | |
| Student | 1.00 (Ref.) | |
| Employee | 0.91 (0.68–1.21) | 0.52 |
| Monthly income (QAR) | | |
| 0–4999 | 1.00 (Ref.) | - |
| 5000–9999 | 1.59 (1.21–2.10) | 0.001 |
| 10000–24999 | 1.44 (1.11–1.87) | 0.006 |
| 25000–39999 | 1.74 (1.26–2.40) | 0.001 |
| 40000+ | 2.94 (2.03–4.27) | <0.0001 |
| Self-rated health | | |
| Fair | 1 (Ref.) | - |
| Good | 0.67 (0.50–0.89) | 0.006 |
| Excellent | 0.39 (0.29–0.53) | <0.0001 |
| Smoking at Home | | |
| No smoker at home | 1.00 (Ref.) | - |
| Smoker at home | 2.21 (1.92–2.54) | <0.0001 |

[*] Factors with p-value < 0.05 were considered statistically significant.

These measures include the recent introduction of tobacco taxes to increase the cigarette prices in 2019, increased anti-smoking health awareness and promotion efforts for the public, and expansion of the tobacco dependence treatment services across Qatar such as in primary health care facilities, hospitals, and specialized cessation centers bearing the full cost of cessation support and treatment for all Qataris and non-Qataris [21]. These measures are likely to have changed the smoking pattern of Qatar's population over the past 19 years.

In the present study, the mean age of tobacco use/smoking onset (19.7 years old) was higher than reported by the GATS in 2013 (18.1 years old) [11], and also higher than in Kuwait (18 years) [22]. This may be reflective of the government's tobacco control measures and efforts implemented to curb smoking. Moreover, 18.8% respondents started using/smoking tobacco before the age of 15 years old. This is much lower than the global figure (80%) [23], however, this necessitates that increased penalty must be imposed on vendors who sell cigarettes to minors. These penalties can include measures like higher fines, revocation of business license, and serving jail time. This also reflects the need to incorporate health education (inclusive of

the harmful effects of smoking) as a subject in school curricula and to encourage healthy leisure time activities to deter smoking and enhance children's skills to resist and refuse smoking. It is worth mentioning that public schools in Qatar have trained personnel who are able to provide smoking cessation counseling to students. This is related to the trainings coordinated by the Tobacco Control Center, Hamad Medical Corporation with the collaboration of international organizations such as WHO and others. The trainings are focused on building capacity, competence and confidence of school counselors and nurses to contribute effectively to the prevention of tobacco use among adolescents and early detection and referral of smokers to the Tobacco Control Center [24].

The most commonly smoked tobacco product in our study was cigarette (42.8%) and waterpipe (20.9%). Cigarettes are also the most consumed tobacco product in other EMR countries [4, 25]. Currently, the tobacco control policies are predominantly focused on cigarettes than other tobacco products. However, the sale of waterpipe also need to be regulated considering the social and cultural acceptability of waterpipe smoking particularly among the youth across the EMR [26, 27]. Concerted efforts are being made in Qatar to prevent the spread of waterpipe smoking, especially among the youth and teenagers. For intance, Al-Mulla et al. conducted a study to test the impact and feasibility of introducing a waterpipe prevention program to control the use of waterpipe among 7th and 8th graders in Doha. This study demonstrated the benefits of creating school-based interventions that can help delay or prevent initiation to waterpipe use [28, 29].

So far there has been no population-based survey reporting the use of *medwakh* in Qatar. In our study, *medwakh* smoking only was more prevalent among Qatari users compared to non-Qatari users (4.7% vs. 2.3%). These rates are lower than the regional rates [30–34], where in United Arab Emirates it is ranked as the second most common form of tobacco [35]. Comparable low rates of smokeless tobacco use only was evident among Qatari and non-Qatari tobacco users (2.9% vs. 1.3%), much lower than the limited data found in the published literature [36, 37].

Electronic cigarette use only was similar among Qatari tobacco users and non-Qatari tobacco users (1.9% vs. 2.0%). As per the Ministry of Public Health revised law in 2016, the selling, distribution or possession of large number of electronic cigarettes is illegal in Qatar. Despite this, there seems to be an increasing trend in the country in comparison to the results of GATS 2013 [11].

The average number of cigarettes smoked by the study respondents was 11.5 cigarettes per day, less than 17.2 cigarettes per day in GATS 2013 [11]. Compared to neighboring countries, this average number of cigarette use is lower than the 18.9 cigarettes per day reported by Kuwait [14], and 23.7 cigarettes per day reported in Iraq [16], but higher than 8.6 cigarettes per day reported in Oman [13]. The decrease in the average number of cigarettes smoked may reflect an actual decrease in the demand or an increase in the price of tobacco products. Over the past year, cigarette prices in the country have risen as an effect of high tobacco taxation set at 100%. The department of taxation in the General Authority of Customs reported that the weight of tobacco import has decreased as a result of this (personal communication, May 2019). Imposing higher taxes on tobacco products may be an effective strategy to reduce consumption of tobacco and boost smoking cessation. Data show that increasing the tax on tobacco products is an avenue for the government to effectively decrease the affordability of cigarettes, potentially curb smoking-related diseases, and make use of the revenues for public services [38–40].

In our mutivaribale model, male respondents were more likely to be tobacco users compared to females (AOR 6.37, 95%, CI 5.36–7.58). Smoking among women is socially unacceptable in Qatar [12], a trend commonly observed in other Arab countries as well [5]. Non-

Qataris were more likely to be current tobacco users compared to Qataris. However, we should interpret this finding with caution, since a higher prevalence rate and dependency of tobacco use was noticed among Qataris as compared to non-Qataris particularly among males. A lower smoking prevalence was found in the older age group categories. This finding could be explained on account of the elderly, more likely, to experience smoking related health problems thus making them more receptive to medical advice and amenable to smoking cessation. Consistent with the literature, higher levels of income, living with a smoker, poor self-rated health increased the likelihood of current tobacco use [12, 41–44]. Currently, the indicators of tobacco consumption in Qatar are better than in the past and in comparison to other high-income countries in the EMR, however, there is potential for improvement in tobacco control.

The tobacco industry aggressively targets adults with variant tobacco product promotions. There is an increasing trend for dual, poly-tobacco, and nonconventional tobacco product use. Our study provides new insights pertaining to the use of specific tobacco product patterns in Qatar. This has not been previously explored in detail, not even in the recent 2013 GATS survey.

Our study findings enhance the evidence base needed to inform health-related programming efforts and other community awareness initiatives needed in Qatar. The study findings can also be used to inform new policy recommendations and offer novel interventions relevant to the local context of Qatar. The rising incidence of waterpipe smoking and multi-tobacco use must be taken into consideration when designing tobacco control programs. To better understand the complex issues surrounding tobacco use in Qatar, further innovative research using rigorous methodology is needed to generate additional data and guide the development of evidence based smoking cessation programs. Also, intervention research is critically needed to determine best practices for reducing tobacco use.

This study has several limitations; principal among them is its cross-sectional design which limits study findings to the reporting of associations between current smoking status and exposure. The strengths of this study include the study design with its large sample size. Moreover, the anonymous self-administered survey technique employed respecting privacy and confidentiality ensuring a high response rate (89.7%). The prevalence results can thus act as a baseline for future studies.

## Conclusion

To our knowledge this large population-based cross-sectional study provides the first evidence for the prevalence of different types of tobacco use including *medwakh* smoking among adults (Qataris and non-Qataris) 18 years and above in Qatar. This study reports lower prevalence of current tobacco use in Qatar (25.2%). This indicates that the introduction of tobacco control laws since 2002 and upon signing the FCTC in 2003, such as prohibition of advertisement on tobacco products and event sponsorship from tobacco companies, control on the sales of tobacco products to minors, designation of smoke-free areas, increase in tobacco taxes and expansions in provision of smoking cessation services have been effective. A comphensive, multisectoral tobacco control strategy which relies on government laws and their enforcement, community awareness aimed at all segments of population including adolescents, and the avialability of smoking cessation programs is vitally important to achieve further progress. Measures such as a ban on tobacco adversting, restrictions on smoking in public places/working environments, tobacco taxation, health warnings against smoking on tobacco products, and other community driven activities are likely to yield additional success and better results.

## Supporting information

**S1 File.**
(DOCX)

## Acknowledgments

The findings achieved herein are solely the responsibility of the authors. We thank the public relations officers and the volunteers in our data collection sites for facilitating the process of recruitment. The authors are also thankful for all the participants for filling out the surveys.

## Author Contributions

**Conceptualization:** Ahmad AlMulla.

**Formal analysis:** Patrick Maisonneuve, Silva Kouyoumjian.

**Funding acquisition:** Ahmad AlMulla.

**Investigation:** Ahmad AlMulla.

**Methodology:** Ahmad AlMulla.

**Project administration:** Jamal Abdullah BaSuhai, Gafar Mahmoud, Silva Kouyoumjian.

**Writing – original draft:** Silva Kouyoumjian.

**Writing – review & editing:** Ahmad AlMulla, Ravinder Mamtani, Sohaila Cheema, Patrick Maisonneuve.

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
