## [Decision Letter · Decision Letter 0]

8 Feb 2021

PONE-D-21-01818

Epidemiology of tobacco use in Qatar: prevalence and its associated factors

PLOS ONE

Dear Dr. Kouyoumjian,

Thank you for submitting your manuscript to PLOS ONE. After careful consideration, we feel that it has merit but does not fully meet PLOS ONE’s publication criteria as it currently stands. Therefore, we invite you to submit a revised version of the manuscript that addresses the points raised during the review process.

The reviewer was correct in pointing out the relatively little additional information added by this study to advance our understanding of tobacco use in Qatar. Although Plos One evaluates studies based on their methodological soundness, method-wise accuracy and rigor are also considered an important value to be added by research. I suggest revise the current study by highlighting how the study has a rigorous study design and employed solid method to assess tobacco use in Qatar.

We look forward to receiving your revised manuscript.

Kind regards,

Xiaozhao Yousef Yang

Academic Editor

PLOS ONE

Journal Requirements:

4.Please include your tables as part of your main manuscript and remove the individual files. Please note that supplementary tables (should remain/ be uploaded) as separate "supporting information" files.

Reviewers' comments:

Reviewer's Responses to Questions

**Comments to the Author**

1. Is the manuscript technically sound, and do the data support the conclusions?

Reviewer #1: Yes

2. Has the statistical analysis been performed appropriately and rigorously? 

Reviewer #1: Yes

3. Have the authors made all data underlying the findings in their manuscript fully available?

Reviewer #1: Yes

4. Is the manuscript presented in an intelligible fashion and written in standard English?

Reviewer #1: Yes

5. Review Comments to the Author

Reviewer #1: This study examined the prevalence of tobacco use and the associated factors among a population-based sample of adults 18 years and above in Qatar. Its sampling methodology, measurement of the variables, and statistical analyses are appropriate. The critical issue is the little insightful information provided to the scholarship.

6. PLOS authors have the option to publish the peer review history of their article (what does this mean?). If published, this will include your full peer review and any attached files.

Reviewer #1: No

---

## [Author Response · Author response to Decision Letter 0]

21 Mar 2021

Epidemiology of tobacco use in Qatar: prevalence and its associated factors

Ahmad AlMulla, 1 Ravinder Mamtani2, Sohaila Cheema2, Patrick Maisonneuve3, Jamal Abdullah BaSuhai1, Gafar Mahmoud1, Silva P. Kouyoumjian1*

1 Tobacco Control Center, WHO Collaborative Center, Department of Medicine, Hamad Medical Corporation, Doha, Qatar. 

2 Institute for Population Health, Weill Cornell Medicine-Qatar, Doha, Qatar.

3 Unit of Clinical Epidemiology, IEO Istituto Europeo di Oncologia IRCSS, Milan, Italy. 

REPLY TO EDITOR AND REVIEWER COMMENTS

All authors would like to thank the editor and the reviewer for assessing our work and for providing us with the constructive feedback that will further improve the article’s quality and its readability. Please find below a point-by-point response addressing the editor/reviewer comments. We have also incorporated these suggestions in the revised manuscript as noted below. 

Note: All references to the revised manuscript pertain to the marked copy of these files including changes implemented through “track changes”.

Editorial requests:

The reviewer was correct in pointing out the relatively little additional information added by this study to advance our understanding of tobacco use in Qatar. Although Plos One evaluates studies based on their methodological soundness, method-wise accuracy and rigor are also considered an important value to be added by research. I suggest revise the current study by highlighting how the study has a rigorous study design and employed solid method to assess tobacco use in Qatar.

Response: We would like to thank the Editor for this suggestion. We have now included this suggestion in the manuscript to highlight the rigorous study design and structured approach to assess tobacco use in Qatar:

The study’s research design and implementation approach were based on a systematic process with due attention given to reduce bias at each stage of the study. The field work was exemplary across all sample recruitment sites to ensure that the results were replicable and valid to support future decision-making for tobacco control. (Materials and Methods, 2nd paragraph, page 5-6).

The survey instrument was carefully administered with attention to protocol adherence by trained staff to guarantee a unified procedure. (Materials and Methods, 3rd paragraph, page 8).

We have revised Table 3 (page 28), which we believe is more readable and informative than before. Accordingly, we have revised the percentages in the text as well:

1. Of the 1550 tobacco users, 42.8% were cigarette smokers, 20.9% waterpipe, 3.2% medwakh (Arabic traditional pipe) and 0.7% cigar. Moreover, 1.9% reported smokeless tobacco use (sweika), 2.0% reported electronic cigarette use, and 0.3% heat-not-burn tobacco use. (Abstract, page 2)

2. Table 3 describes the number and percentage of different types of tobacco among current users by nationality. The numbers of different types of tobacco use are not exclusive and further details are found in S6-S9 Tables. Of the current tobacco users (n=1550), 42.8% were cigarette smokers, 20.9% waterpipe, 3.2% medwakh (Arabic traditional pipe) and 0.7% cigar smokers. Moreover, 1.9% reported smokeless tobacco use (sweika), 2.0% electronic cigarette use, 0.3% heat-not-burn tobacco use and 28.1% were using more than one type of tobacco product (Table 3). (Results, 2nd paragraph, page 11)

3. Among Qatari current tobacco users (n=617), 36.3% were cigarette smokers, followed by waterpipe smoking (18.2%), medwakh smoking (4.7%) and cigar smoking (0.5%). Moreover, 2.9% reported smokeless tobacco use (sweika), 1.9% electronic cigarette use, 0.6% heat-not-burn tobacco use and 34.8% were using more than one type of tobacco product (Table 3). (Results, 3rd paragraph, page 11)

4. Among non-Qatari tobacco users (n=912), cigarette smoking was the highest (47.6%), followed by waterpipe smoking (22.5%), medwakh smoking (2.3%) and cigar smoking (0.9%). Moreover, 1.3% reported smokeless tobacco use (sweika), electronic cigarette use (2.0%), and 0.1% heat-not-burn tobacco use, and 23.4% were using more than one type of tobacco product (Table 3). (Results, 1st paragraph, page 12)

5. The most commonly smoked tobacco product in our study was cigarette (42.8%) and waterpipe (20.9%). (Discussion, last paragraph, page 15) 

6. So far there has been no population-based survey reporting the use of medwakh in Qatar. In our study, medwakh smoking only was more prevalent among Qatari users compared to non-Qatari users (4.7% vs. 2.3%). These rates are lower than the regional rates (30-34), where in United Arab Emirates it is ranked as the second most common form of tobacco (35). Comparable low rates of smokeless tobacco use only was evident among Qatari and non-Qatari tobacco users (2.9% vs. 1.3%), much lower than the limited data found in the published literature (36, 37). (Discussion, 1st paragraph, page 16)

7. Electronic cigarette use only was similar among Qatari tobacco users and non-Qatari tobacco users (1.9% vs. 2.0%). As per the Ministry of Public Health revised law in 2016, the selling, distribution or possession of large number of electronic cigarettes is illegal in Qatar. Despite this, there seems to be an increasing trend in the country in comparison to the results of GATS 2013 (11). (Discussion, 2nd paragraph, page 16)

8. A typo error in Table 2 has been corrected as well. (Table 2, page 26)

Response: We confirm that the manuscript meets PLOS ONE’s style requirements including those for file naming.

Response: As per the suggestion, a title page has been included into the beginning of the manuscript file itself, listing all the authors and their affiliations.

Response: Thank you for this comment. The survey tool is available in Arabic and English and is provided as Supporting Information (S1 Appendix and S2 Appendix of SI). (Materials and Methods, page 7, 2nd paragraph).

4.Please include your tables as part of your main manuscript and remove the individual files. Please note that supplementary tables (should remain/ be uploaded) as separate "supporting information" files.

Response: The tables have now been included as part of the main manuscript. (Main document_marked copy, pages 25-30)

Response: Captions for the Supporting Information files have been included at the end of the manuscript on page 23 and 24 according to PLOS ONE guidelines.

Response: The ethics statement i.e “The study protocol was approved by the Institutional Review Board, Medical Research Center, Hamad Medical Corporation, Qatar” now appears only in the Methods section of the manuscript. (Materials and Methods, page 6, 1st paragraph).

Reviewers' comments to the author:

This study examined the prevalence of tobacco use and the associated factors among a population-based sample of adults 18 years and above in Qatar. Its sampling methodology, measurement of the variables, and statistical analyses are appropriate. The critical issue is the little insightful information provided to the scholarship.

Response: We thank the reviewer for this comment. Our study now provides insightful information for scholarship as follows (Discussion, 2nd paragraph, page 17 and 1st paragraph, page 18):

The prevalence of tobacco smoking in Qatar has reduced by 15.2% between 2000 and 2019 due to the measures implemented by the country. Our study strengthens the previously limited tobacco evidence available for Qatar. Recommendations and way forward are also discussed. 

1) Moreover, the tobacco industry aggressively targets adults with variant tobacco product promotions. There is an increasing trend for dual, poly-tobacco, and nonconventional tobacco product use. Our study provides new insights pertaining to the use of specific tobacco product patterns in Qatar. This has not been previously explored in detail, not even in the recent 2013 GATS survey. 

2) Our study findings enhance the evidence base needed to inform health-related programming efforts and other community awareness initiatives needed in Qatar. 

3) The study findings can also be used to inform new policy recommendations and offer novel interventions relevant to the local context of Qatar. The rising incidence of waterpipe smoking and multi-tobacco use must be taken into consideration when designing tobacco control programs. 

4) To better understand the complex issues surrounding tobacco use in Qatar, further innovative research using rigorous methodology is needed to generate additional data and guide the development of evidence based smoking cessation programs. Also, intervention research is critically needed to determine best practices for reducing tobacco use.

---

## [Decision Letter · Decision Letter 1]

31 Mar 2021

Epidemiology of tobacco use in Qatar: prevalence and its associated factors

PONE-D-21-01818R1

Dear Dr. Kouyoumjian,

We’re pleased to inform you that your manuscript has been judged scientifically suitable for publication and will be formally accepted for publication once it meets all outstanding technical requirements.

Kind regards,

Xiaozhao Yousef Yang, Ph.D.

Academic Editor

PLOS ONE

Additional Editor Comments (optional):

Reviewers' comments:

Reviewer's Responses to Questions

**Comments to the Author**

1. If the authors have adequately addressed your comments raised in a previous round of review and you feel that this manuscript is now acceptable for publication, you may indicate that here to bypass the “Comments to the Author” section, enter your conflict of interest statement in the “Confidential to Editor” section, and submit your "Accept" recommendation.

Reviewer #1: All comments have been addressed

2. Is the manuscript technically sound, and do the data support the conclusions?

Reviewer #1: Yes

3. Has the statistical analysis been performed appropriately and rigorously? 

Reviewer #1: Yes

4. Have the authors made all data underlying the findings in their manuscript fully available?

Reviewer #1: Yes

5. Is the manuscript presented in an intelligible fashion and written in standard English?

Reviewer #1: Yes

6. Review Comments to the Author

Reviewer #1: The authors have adequately addressed my comments, this paper has great improved though revising. I think this paper may be published .

7. PLOS authors have the option to publish the peer review history of their article (what does this mean?). If published, this will include your full peer review and any attached files.

Reviewer #1: **Yes: **Tingzhong Yang

---

## [Editor Report · Acceptance letter]

6 Apr 2021

PONE-D-21-01818R1 

Epidemiology of tobacco use in Qatar: prevalence and its associated factors 

Dear Dr. Kouyoumjian:

I'm pleased to inform you that your manuscript has been deemed suitable for publication in PLOS ONE. Congratulations! Your manuscript is now with our production department. 

Kind regards, 

on behalf of

Dr. Xiaozhao Yousef Yang 

Academic Editor

PLOS ONE